# Patterns of SARS-CoV-2 seropositivity among essential workers in long term care and retirement homes in Ontario, Canada: A descriptive cross-sectional study

**Christine Fahim**[1,2*], **Siyi Wang**[1], **Nimitha Paul**[1], **Karen Colwill**[3], **Roya Dayam**[3], **Jamie M. Boyd**[1], **Huiting Ma**[1], **Vincenza Gruppuso**[1], **Ana Mrazovac**[1], **Jessica Firman**[1], **Anjali Patel**[1], **Vanessa Bach**[1], **Keelia Quinn de Launay**[1], **Alyson Takaoka**[1], **Vanja Grubac**[1], **Anne-Claude Gingras**[2,3], **Sharon E. Straus**[1,2,4], **Sharmistha Mishra**[1,4,5,6,7]

**1** Li Ka Shing Knowledge Institute, Unity Health Toronto, Toronto, Canada, **2** Dalla Lana School of Public Health, University of Toronto, Toronto, Canada, **3** Lunenfeld-Tanenbaum Research Institute, Sinai Health, Toronto, Canada, **4** Department of Medicine, University of Toronto, Toronto, Canada, **5** Department of Molecular Genetics, University of Toronto, Toronto, Canada, **6** Institute of Medical Sciences, University of Toronto, Toronto, Canada, **7** Institute of Clinical and Evaluative Sciences, Toronto, Canada

\* christine.fahim@unityhealth.to

## Abstract

Understanding patterns of SARS-CoV-2 seroprevalence among Long-Term Care Home and Retirement Home (LTCH/RH) staff is critical to designing effective public health interventions. We estimated SARS-CoV-2 seroprevalence among LTCH/RH staff in Ontario, Canada between May 2021-October 2022 using a cross-sectional analysis. Eligible participants completed a demographic questionnaire and provided a dried blood spot sample. Positive seroprevalence was defined as the proportion of individuals in a population who were positive for a SARS-CoV-2 infection, determined using anti-nucleocapsid total IgG antibodies analyzed with a validated chemiluminescent ELISA. We report age-adjusted prevalence ratios [PR; confidence interval, CI] by participant socio-demographic, household, neighbourhood, and occupational characteristics and stratified the analyses over two time periods (period 1: 2021-05-17 to 2021-12-31; period 2: 2022-01-02 to 2022-10-25). A total of 603 staff were included in our analysis; n=235 (39%) were enrolled in period 1 and n=368 (61%) were enrolled in period 2. Seroprevalence was 24% and 44% in periods 1 and 2, respectively. Age-adjusted prevalence ratios were nearly 2-fold higher among Black [PR 1.78; CI 1.28-2.48], East and Southeast Asian [PR 1.55, CI 1.18-2.04] and other racialized participants [PR 1.42, CI 1.03-1.96] compared to White participants. We did not observe a pattern across household characteristics, although we observed a trend towards higher seropositivity among participants living in COVID-19 hotspots. Prevalence ratios were lower for participants in higher income neighbourhoods [PR 0.72, CI 0.58-0.98]. We did not observe variability in seroprevalence across occupational characteristics with the exception of paid sick leave which was higher among participants with home-provided paid sick leave at the time of the survey [PR 0.58, CI 0.45-0.75]. Among LTCH/RH staff, we found important sources of variability of SARS-CoV-2 seroprevalence and

**Data availability statement:** The study data used in this manuscript have been de-identified and made available via Open Science Framework. Data can be accessed at: https://osf.io/zt7ux.

**Funding:** This study was funded by the COVID-19 Immunity Task Force via the Public Health Agency of Canada (#2021-HQ-000143; grant awarded to SES). The funders had no role in study design, data collection and analysis, decision to publish, or preparation of the manuscript. ). SM (CRC number 950-232643) is supported by a Canada Research Chair (Tier 2) in Mathematical Modeling and Program Science. SES holds a Canada Research Chair (Tier 1) in in Knowledge Translation and Quality of Care. ACG holds a Canada Research Chair (Tier 1) in Functional Proteomics.

**Competing interests:** The authors have declared that no competing interests exist.

**Abbreviations:** CI, Confidence Interval; DBS, Dried Blood Spot; ELISA, Enzyme-linked immunoassay; LTCH, Long Term Care Homes; RH, Retirement Homes; PR, Prevalence ratio (age-adjusted)

strong correlations with socioeconomic disparities. Our findings show the importance of designing equity-rooted health interventions that recognize the intersection between community and the workplace.

## Introduction

Staff working in LTCH/RH represent a diversity of occupational and socio-demographic characteristics and experiences which could shape exposure risks as they relate to SARS-CoV-2 acquisition. Occupations in LTCH/RH range from regulated and higher-wage positions (e.g., physicians, nurse practitioners, and management) to unregulated and lower-wage positions such as personal support workers or care aides [1]. Unregulated workers do not have mandatory regulations, training, or registration, often resulting in decreased job protections, benefits and pay [2]. It is estimated that personal support workers and care aides provide 90% of the direct, person-to-person, care for residents in LTCH/RH [3–5].

In Ontario, Canada, nearly half of all personal support workers are employed in LTCH [6]. Recent data show that approximately 90% of personal support workers in the Greater Toronto Area identify as racially minoritized, and 55% report living in lower income households [6]. Less is known about the remainder of the LTCH/RH workforce. The aforementioned studies have made important contributions to understanding the experiences of subsets of LTCH staff, but none to date have characterized occupational and household characteristics across the range of the LTCH and RH workforce.

An extensive body of work on the SARS-CoV-2 epidemic in Canada has documented the complex systemic and multisectoral factors that shaped social determinants of health, including occupational health and household characteristics [7–9]. These factors have been shown to drive social and physical networks and workplace safety and security (e.g., transportation to/from onsite work, infection control resources, services, and institutional policies and programs), which in turn, collectively shape SARS-CoV-2 acquisition and transmission risks among essential workers. Yet, to date, there has not been a description of the pattern of SARS-CoV-2 risks, measured using seroprevalence, among LTCH and RH staff in Canada, particularly after the early waves of the epidemic in 2020 and after vaccine roll-out [10].

We therefore sought to characterise patterns of seroprevalence among LTCH/RH workers in Ontario between May 2021 and October 2022, after vaccine roll-out; and across socio-demographic, household, neighbourhood and occupational characteristics.

## Methods

### Study design and period

We conducted an observational, cross-sectional study using baseline data from the Wellness Hub study. The Wellness Hub study was designed to measure seroprevalence of SARS-CoV-2 antibodies among LTCH and RH staff, residents, and caregivers. It is a prospective cohort study conducted among LTCH/RH populations in Ontario, Canada recruited and enrolled into the cohort between May 17, 2021 and October 25, 2022. In this paper, we use the baseline data from LTCH/RH staff cohort, and restricted to participants residing in the Greater Toronto Area and its surrounding areas (e.g., Hamilton). We report our study findings using the Strengthening the Reporting of Observational Studies in Epidemiology checklist [11].

### Sampling frame and sampling strategy

Wellness Hub study sites were recruited using both purposive and convenience sampling. Recruitment strategies included both active (e.g., tailored presentations to LTCH/RH

networks) and passive (e.g., website/newsletter advertisements, word of mouth) strategies. We leveraged an extensive network of project partners to disseminate the study materials and facilitate recruitment (S1 Text). Within recruited homes, we recruited staff using convenience and snowball sampling. Recruitment strategies included: emails and phone calls to LTCH/RH managers and staff, advertisements via posters, newsletters, on-site recruitment days, social media, and study self-enrollment via website URLs and QR codes.

## Setting

We recruited 72 LTCH/RH to participate in the Wellness Hub study. Homes were eligible to participate in Wellness Hub if they agreed to participate in individual-level recruitment, a home-level needs assessment interview to address challenges facing their home during the COVID-19 pandemic and if they identified a designated 'point person' to act as a primary contact for the study. Indigenous LTCH/RH and Indigenous participants were excluded from the study, given resource and experience gaps on our study team to appropriately engage members of Indigenous communities. Within all participating homes, four LTCH/RH populations (staff, household members of staff, residents, and essential care partners of residents [e.g., family, caregivers]) were eligible to participate in the study by providing DBS samples and completing an in-depth questionnaire; in this article, we present the findings from the LTCH/RH staff cohort.

## Participants

Participants were eligible for inclusion if they: identified as a staff (including full-time, part-time paid staff and unpaid volunteers) working at a LTCH/RH at the time of consent and enrollment; were comfortable speaking/reading English or French; were at least 18 years old; were willing to complete a study demographic questionnaire and provide a DBS sample.

## Data collection

Participants completed a demographic questionnaire (S2 Text) and provided a DBS sample. Demographic questionnaires included multiple choice and open-ended questions, and were completed using one of the following methods: 1) a hard-copy printed version of the questionnaire, 2) an online version of the questionnaire via REDCap platform or 3) a questionnaire completed through an in-person or virtual call led by a research team member. Participants provided DBS samples via one of the following methods: 1) collected in-person with the support of study staff or LTCH/RH staff, or 2) self-collected by participants following detailed instructions. Participants had the option of leaving their self-collected sample at a drop-off site within the study home or mailing the sample back to the study team.

## DBS sample storage and testing

DBS were collected on Whatman 903 cards using a blade lancet for finger pricking. No identifying information was listed on the cards, except a unique participant ID. Collection of ≥2 filled spots on the card was requested. Cards were then air dried (according to the manufacturer's instructions) and placed in individual sealable bags, along with a desiccant pouch. Cards were transferred to the Gingras laboratory at Sinai Health. Upon reception in that lab, the cards were placed at 4°C for short-term storage, or frozen (at -80°C) for long-term storage. Cards were then brought to room temperature before the packages were opened, and a portion of the spot (2 x 3 mm or a 1 x 3 mm diameter circle if the sample was limited as this was sufficient for one pass at the assays if the blood soaked through the Whatman paper) was then extracted with a semi-automated hole puncher directly into a 96-well plate.

## Outcome measure

Our primary outcome was seropositivity, defined as having a SARS-CoV-2 infection. SARS-CoV-2 infection status was determined using DBS, based on anti-nucleocapsid total IgG antibodies by a validated chemiluminescent ELISA. In alignment with another seroprevalence study conducted in LTCH during this time period [12], we performed our analyses at a specificity threshold of 90% (a relative ratio to a synthetic standard of 0.341; 11.34 binding antibody units [BAU/mL], corresponding to 88% sensitivity in a vaccinated cohort, which provides the best predictive value (negative predictive value * positive predictive value) when seroprevalence is 40% or greater [13]. As noted in several studies, many individuals who have breakthrough infection after vaccination either do not seroconvert for nucleocapsid or had lower antibody levels to it [14]. Thus, we used the relative ratio nucleocapsid threshold of 0.341 (BAU of 11.34 BAU/mL) to reduce false negatives. DBS passing the threshold of seropositivity for anti-nucleocapsid IgG antibodies were identified as having a natural COVID-19 infection and were marked as 'seropositive for natural infection'; DBS that did not pass the threshold were considered 'seronegative for natural infection' (see S3 Text for more detail).

## Conceptual framework and variables

Our conceptual framework in selecting variables to examine patterns of seroprevalence was informed by literature on epidemiological and socioeconomic factors associated heightened risks of SARS-CoV-2 acquisition in Canada; and literature more specific to exposure risks in occupational settings [9,15–17]. We classified variables of interest according to four domains conceptualized based on potential contexts that shape exposure risks for SARS-CoV-2 among LTCH/RH staff: sociodemographic, household characteristics, neighbourhood characteristics, and occupational characteristics. Sociodemographic characteristics included: age [18–20], gender [21], educational attainment [22]; and race as a proxy for individual, institutional, and structural racism [9,22–24]. Household characteristics included: household income [8], household density [9,22], and housing type [22,25,26] (S1 Table). Neighbourhood-level characteristics refer to the residential neighbourhood of the participant (not the LTCH/RH) and include: socioeconomic indicators of neighbourhood-level median household income (measured as tertile) and proportion essential workers (measured as tertile). The tertiles for neighbourhood-level median household income and proportion essential workers were based on data at the level of the dissemination area from the 2016 Census from Statistics Canada [27], with details included in S4 Text. We also included a neighborhood-level measure of community transmission, defined as "hotspot versus non-hotspot" using the neighbourhood-level cumulative SARS-CoV-2 diagnoses. We used the first three digits of participant postal code of residence (called the forward sortation area) to create the neighbourhood-level hotspot variable as a dichotomous measure. A hotspot referred to neighbourhoods (defined using the forward sortation area) in Ontario that comprised 20% of the population with the highest per-capita incidence of diagnosed SARS-CoV-2 cases between January 23, 2020, and January 16, 2021 [28,29]. We defined the remaining geographical areas as non-hotspots. S4 Text details the derivation of neighbourhood-level variables including hotspots, dissemination area characteristics, and forward sortation area levels.

Finally, our last domain centered on occupational-related characteristics, and included: occupational role, employment status, type of home (LTCH or RH), transportation used to attend the home for work, and current paid sick leave. We classified occupational role using the self-reported answer to "what best describes your job title at this LTCH/RH" and regrouped open text responses based on best fit to the categories. We classified employment status using the self-reported answer to: "what is your employment status in this LTCH/RH"

with open text responses coded into existing categories. We grouped modes of transportation to the home based on anticipated magnitude of contacts during occupation-related movement from/to households from workplaces: few contacts (i.e., working from household or remote work) to greater number of contacts (i.e., use of rideshares or carpooling; use of public transportation). The variable of current paid sick leave at the home of employment was generated using self-reported answer to: "do you have paid sick leave at this facility". If individuals answered "no", "do not know/ unsure", or if responses were missing, we categorized their response as "no" for current paid sick leave.

## Analysis plan

Survey data (S2 Text) with responses fewer than n=5 were collapsed in order to protect participant anonymity. Open-ended responses were independently themed by a Research Scientist (CF) and Research Coordinator (NP) to facilitate analysis. Continuous data were reported using means, standard deviations (SD) and medians, and inter-quartile range (IQR). We conducted descriptive analyses and reported seropositivity (i.e., seroprevalence) by the covariates of interest. We then generated age-adjusted prevalence ratios (PR) with 95% confidence intervals (95% CI) for each covariate using normal approximation (Wald), and using *a priori* selection of the reference category as appropriate. For age-adjustment, we applied logistic regression with the log link function for the PR, where the outcome was seroprevalence and included the age as a continuous variable. Data cleaning, analyses, and visualization were conducted in R 4.3.0. We used the "stats" and "crosstable" packages to conduct our analysis (Table 1) and the "ggplot2" package to generate Fig 1.

## Research ethics

This study was approved by the Unity Health Toronto Research Ethics Board (REB 20-347). Participants provided formal consent between May 6 2021 and June 26 2023. All participants or their shared decision maker provided either written (hard copy) or online (captured via REDCap) consent. There were no minors involved in the study (participant age range 18-79 years).

# Results

## Participant characteristics

Of the n=797 LTCH/RH staff who participated in the Wellness Hub study and completed a demographic questionnaire, n=603 provided a DBS and were included in this analysis. Among the 603 participants, 235 (39%) were enrolled in period 1 and 368 (61%) in period 2. Nearly all (97.7%) participants reported receipt of at least 2 doses or equivalent of SARS-CoV-2 vaccination (37.1% of participants in period 1 and 60.8% in period 2). Table 1 summarizes participant characteristics under the conceptual framework. Eighty-three percent of the participants identified as women, 12% as men, and 4% chose not to specify. Nearly half (44%) of participants identified as White, 24% as East or Southeast Asian, 10% as Black, and 16% identified as a member of a non-Black, non-East/Southeast Asian racialized community (Table 1). The crude SARS-CoV-2 seroprevalence in our sample was 24% in period 1 and 44% in period 2 (Table 1, S2 Table).

## Age-adjusted pattern of seroprevalence

Seroprevalence was nearly 2-fold higher in period 2 compared to period 1 [PR 1.87; CI 1.44-2.43]. Seropositivity was nearly 2-fold higher among racialized participants compared with

**Table 1. Demographic Characteristics.**

| Group | Label | Variable | Negative - N=386 (column proportion) | Positive - N=217 (column proportion) | Total - N=603 (column proportion) | Seroprevalence (Positive/ Total) |
|---|---|---|---|---|---|---|
| Period | Time period (before/ after 2022-01-01) | First period | 179 (46%) | 56 (26%) | 235 (39%) | 24% |
| | | Second period | 207 (54%) | 161 (74%) | 368 (61%) | 44% |
| Socio-demographic characteristics | Age | Min/ Max | 18.0/ 97.0 | 18.0/ 74.0 | 18.0/ 97.0 | |
| | | Med [IQR] | 47.0 [34.0;56.0] | 47.0 [36.0;55.0] | 47.0 [35.0;56.0] | |
| | | Mean (std) | 45.5 (13.8) | 45.6 (12.4) | 45.5 (13.3) | |
| | | N (exclude missing/unknown) | 371 | 209 | 580 | 36% |
| | | Missing/Unknown | 15 | 8 | 23 | 35% |
| | Age group | 18-24 | 32 (8%) | 9 (4%) | 41 (7%) | 22% |
| | | 25-34 | 62 (16%) | 39 (18%) | 101 (17%) | 39% |
| | | 35-44 | 63 (16%) | 43 (20%) | 106 (18%) | 41% |
| | | 45-54 | 106 (27%) | 62 (29%) | 168 (28%) | 37% |
| | | 55-65 | 89 (23%) | 47 (22%) | 136 (23%) | 35% |
| | | 65+ | 19 (5%) | 9 (4%) | 28 (5%) | 32% |
| | | Missing/unknown | 15 (4%) | 8 (4%) | 23 (4%) | 35% |
| | Gender | Man | 57 (15%) | 18 (8%) | 75 (12%) | 24% |
| | | Woman | 315 (82%) | 186 (86%) | 501 (83%) | 37% |
| | | Missing/unknown | 14 (4%) | 13 (6%) | 27 (4%) | 48% |
| | Educational level | Up to high school graduation | 66 (17%) | 34 (16%) | 100 (17%) | 34% |
| | | College degree or trades certificate | 118 (31%) | 62 (29%) | 180 (30%) | 34% |
| | | University bachelor's degree | 113 (29%) | 74 (34%) | 187 (31%) | 40% |
| | | Graduate or professional degree | 73 (19%) | 34 (16%) | 107 (18%) | 32% |
| | | Missing/unknown | 16 (4%) | 13 (6%) | 29 (5%) | 45% |
| | Race | White | 190 (49%) | 77 (35%) | 267 (44%) | 29% |
| | | Black | 31 (8%) | 30 (14%) | 61 (10%) | 49% |
| | | East or Southeast Asian | 84 (22%) | 63 (29%) | 147 (24%) | 43% |
| | | Other racialized | 60 (16%) | 37 (17%) | 97 (16%) | 38% |
| | | Missing/unknown | 21 (5%) | 10 (5%) | 31 (5%) | 32% |
| Household characteristics | Income (household level) | $0 - $59,999 | 77 (20%) | 42 (19%) | 119 (20%) | 35% |
| | | $60,000 - $89,999 | 60 (16%) | 35 (16%) | 95 (16%) | 37% |
| | | $90,000 or more | 121 (31%) | 77 (35%) | 198 (33%) | 39% |
| | | Missing/unknown | 128 (33%) | 63 (29%) | 191 (32%) | 33% |
| | Number of people in household | 1 | 30 (8%) | 14 (6%) | 44 (7%) | 32% |
| | | 2-4 | 261 (68%) | 146 (67%) | 407 (67%) | 36% |
| | | 5+ | 73 (19%) | 42 (19%) | 115 (19%) | 37% |
| | | Missing/unknown | 22 (6%) | 15 (7%) | 37 (6%) | 41% |
| | Housing Type | Apartment/condo | 86 (22%) | 57 (26%) | 143 (24%) | 40% |
| | | House | 253 (66%) | 136 (63%) | 389 (65%) | 35% |
| | | Other | 18 (5%) | 9 (4%) | 27 (4%) | 33% |
| | | Missing/unknown | 29 (8%) | 15 (7%) | 44 (7%) | 34% |

*(Continued)*

White participants: the prevalence ratio for seropositivity among Black participants was 1.78 [CI 1.28-2.48], 1.55 [CI 1.18-2.04] among East/Southeast Asian participants, and 1.42 [CI 1.03-1.96] among other racialized participants. Although there were few men in our study,

**Table 1.** (Continued)

| Group | Label | Variable | Negative - N=386 (column proportion) | Positive - N=217 (column proportion) | Total - N=603 (column proportion) | Seroprevalence (Positive/ Total) |
|---|---|---|---|---|---|---|
| Neighbourhood-level characteristics | FSA-level hotspot/ non-hotspot indicator | Non-hotspot | 247 (64%) | 121 (56%) | 368 (61%) | 33% |
| | | Hotspot | 115 (30%) | 81 (37%) | 196 (33%) | 41% |
| | | Missing/unknown | 24 (6%) | 15 (7%) | 39 (6%) | 38% |
| | DA-level income tertile | 1 | 116 (30%) | 79 (36%) | 195 (32%) | 41% |
| | | 2 | 123 (32%) | 68 (31%) | 191 (32%) | 36% |
| | | 3 | 118 (31%) | 49 (23%) | 167 (28%) | 29% |
| | | Missing/unknown | 29 (8%) | 21 (10%) | 50 (8%) | 42% |
| | DA-level essential worker tertile | 1 | 107 (28%) | 48 (22%) | 155 (26%) | 31% |
| | | 2 | 132 (34%) | 73 (34%) | 205 (34%) | 36% |
| | | 3 | 118 (31%) | 75 (35%) | 193 (32%) | 39% |
| | | Missing/unknown | 29 (8%) | 21 (10%) | 50 (8%) | 42% |
| Occupational characteristics | Occupation (title) | Administration or management | 83 (22%) | 44 (20%) | 127 (21%) | 35% |
| | | Personal support worker or support staff (e.g.e.g., kitchen, housekeeping, laundry) | 115 (30%) | 74 (34%) | 189 (31%) | 39% |
| | | Physician, nurse or registered practical nurses | 91 (24%) | 55 (25%) | 146 (24%) | 38% |
| | | Other | 62 (16%) | 27 (12%) | 89 (15%) | 30% |
| | | Missing/unknown | 35 (9%) | 17 (8%) | 52 (9%) | 33% |
| | Employment status | Full-time | 252 (65%) | 141 (65%) | 393 (65%) | 36% |
| | | Part-time or agency/contract | 99 (26%) | 54 (25%) | 153 (25%) | 35% |
| | | Other | 16 (4%) | 8 (4%) | 24 (4%) | 33% |
| | | Missing/unknown | 19 (5%) | 14 (6%) | 33 (5%) | 42% |
| | Home Type (LTCH/ RH) | Long-term care home | 298 (77%) | 173 (80%) | 471 (78%) | 37% |
| | | Retirement home | 88 (23%) | 44 (20%) | 132 (22%) | 33% |
| | Transportation to work | Drive alone | 235 (61%) | 127 (59%) | 362 (60%) | 35% |
| | | Walk or cycle | 21 (5%) | 8 (4%) | 29 (5%) | 28% |
| | | Rideshare or carpool (including taxi or uber) | 49 (13%) | 22 (10%) | 71 (12%) | 31% |
| | | Public transport | 58 (15%) | 41 (19%) | 99 (16%) | 41% |
| | | Work from home, or not currently working | 7 (2%) | 6 (3%) | 13 (2%) | 46% |
| | | Missing/unknown | 16 (4%) | 13 (6%) | 29 (5%) | 45% |
| | Paid Sick Leave | Yes | 143 (37%) | 120 (55%) | 263 (44%) | 46% |
| | | No | 185 (48%) | 64 (29%) | 249 (41%) | 26% |
| | | Missing/unknown | 58 (15%) | 33 (15%) | 91 (15%) | 36% |

there was a trend towards a 1.5-fold higher seroprevalence among women [PR 1.48, CI 0.97-2.25] and similar seroprevalence by levels educational attainment (S2 Table).

We did not observe a pattern across household characteristics. There was a meaningful trend towards higher seropositivity among participants residing in hotspots compared with non-hotspots [PR 1.25, CI 1.00-1.57]. Prevalence ratios were lower for participants living in higher income neighbourhoods [PR 0.72, CI 0.58 – 0.98].

We did not observe variability in seroprevalence across occupational characteristics, with the exception of paid sick leave. Seropositvity was higher among participants with paid sick via the home at the time of the survey [PR 0.58, CI 0.45-0.75].

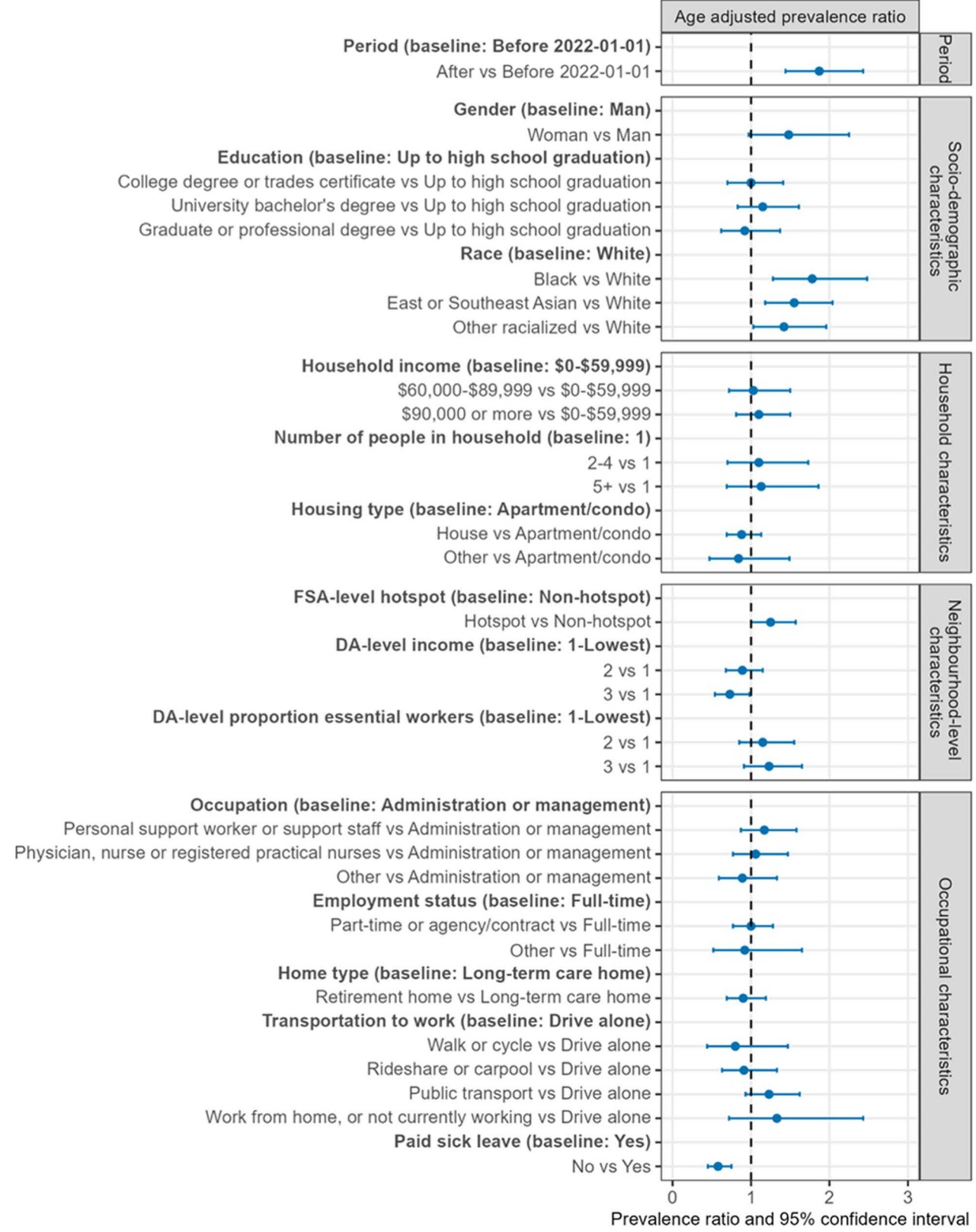

**Fig 1.  Age-adjusted prevalence ratios.**

## Discussion

Baseline data from a cohort study of LTCH and RH staff in Toronto, Canada demonstrated that SARS-CoV-2 seroprevalence was 24% between May-December 2021 and 44% between January-October 2022. We documented racial disparities, with age-adjusted prevalence nearly two-fold higher among Black and East/Southeast Asian staff as compared with White staff. Seroprevalence was also higher among staff living in COVID-19 hotspots. We did not observe variability by household or occupational factors, except for paid sick leave; staff with higher seroprevalence were more likely to report access to paid sick leave.

The patterns of seroprevalence among LTCH/RH staff are reflective of underlying disparities that have been extensively documented in Canada. Racially minoritized communities experienced some of the highest rates of SARS-CoV-2 infections due to disproportionate exposure risks, including through occupational risks. In addition to the burnout, fatigue, and stress often experienced by LTCH/RH staff, racialized staff further contend with racism and discrimination in the workplace. A 2009 Ontario survey with PSWs showed that over half of PSWs experience harassment or workplace violence; 15% of PSWs experienced violence and harassment related to their race [4]; other studies have provided much greater estimates of racism and workplace harassment suggesting these experiences are likely underreported [30]. Within long-term care, racialized workers are overrepresented as frontline caregivers and hold fewer management and administrative roles. The nature of unregulated, precarious work coupled with power imbalances and occupational hierarchies provide limited protections or options [30] for staff, particularly those experiencing racism or discrimination in the workplace [31].

Seroprevalence was higher among staff with paid sick leave. Although studies are not yet available in Canada, the provision of emergency sick leave in the United States was associated with reduced SARS-CoV-2 transmission at a community-level [32]. Access to sick leave has also been shown to reduce presenteeism with other respiratory illnesses [32]. Our finding of higher seroprevalence among staff who had paid sick leave provided by the LTCH, may reflect the fact that sick-leave represents high risks of SARS-CoV-2 exposures, and we did not examine confounders in our descriptive study of patterns. Furthermore, individuals with sick leave may not necessarily be in a position (financially, or other reasons including power hierarchies in occupations such as health care to take sick leave when needed [30]. Of note however, paid sick-leave is mostly relevant for onward transmission in the LTCH, not for acquisition at the individual-level; and our measures of seroprevalence reflect acquisition. Our study was not designed to examine whether sick-leave was associated with lower level of transmission; if most acquisition among staff occur in the household then sick-leave would not necessarily be associated with seroprevalence. A better measure would be to examine association between seropositivity and the proportion of staff in the LTCH who have paid sick-leave, but this measure was not available in our study.

The neighborhood-level correlation between staff seroprevalence and SARS-CoV-2 hotspots signal the importance of community-level factors and community-level transmission when considering exposure risks among LTCH/RH workers. An earlier study in Ontario found that the spatial distribution of SARS-CoV-2 cases among LTCH workers were very closely aligned with the overall community-level distribution of cases [33]. Studies from the United States also suggest that LTCH workers may have been more likely to live in neighborhoods with higher rates of SARS-CoV-2 [34]. Taken together, the implication of these findings is that reducing community-level transmission is an important component of reducing risks among LTCH/RH staff and outbreaks in LTCH/RH. For example, community-level vaccination programs and other efforts to reduce community-level transmission (testing, isolation support) prioritized and tailored to SARS-CoV-2 geographical hotspots simultaneously with prioritization of workers within LTCH/RH could potentially be more effective than focusing on homes in isolation of disparities that connect community-transmission and

workplace exposure risks. Another implication is that community transmission risks need to be accounted for when evaluating the potential nosocomial transmission impact of policies instituted at the LTCH/RH level. Thus, our findings confirm previous research and support the need for policies, programs, and pandemic preparedness and response frameworks that are designed to address unmet prevention needs at the intersections between community and the workplace [33,34]. Although focused on SARS-CoV-2, the policy implications are also relevant for seasonal influenza and other respiratory virus outbreaks and epidemics [35].

### Limitations

Our study is limited to staff who worked in homes that participated in the Wellness Hub study, and geographic scope was restricted to the Greater Toronto Area (a largely urban setting, with a larger proportion of racialized population, and where SARS-CoV-2 transmission during the study period was also higher than in more rural settings). As such, our findings may not be generalizable to less urban settings. We were also limited by grouping our examination of associations across wide ranges of time because participants were recruited over 17 months. Another limitation is measurement bias in our outcome of interest, as we did not have data on polymerase-chain-reaction SARS-CoV-2 testing data to confirm and validate the results from the DBS samples. We may have underestimated past infections because some vaccinated individuals may not have produced sufficient anti-nucleocapsid antibodies to pass the positivity threshold or their antibody levels may have waned below the level of detection at the timing of testing. Finally, the results of our descriptive analyses should be interpreted as observations around variability in patterns of seroprevalence; future study is needed to conduct risk factor and potentially causal analyses.

### Conclusions

In conclusion, we identified important sources of variability in SARS-CoV-2 seroprevalence among LTCH/RH staff in Ontario correlated with staff race/ethnicity, neighbourhood, and socioeconomic status. Age-adjusted prevalence ratios were 2-fold higher among Black, East Asian, SouthEast Asian and other racialized participants compared to White participants. Prevalence ratios were also lower among participants in higher income neighbourhoods and a trend towards higher seropositivity in COVID-19 hotspots was observed.

### Supporting information

**S1 Text. Study Partners.**
(DOCX)

**S2 Text. Demographic questionnaire.**
(DOCX)

**S3 Text. DBS Interpretation.**
(DOCX)

**S4 Text. Study Variables.**
(DOCX)

**S1 Table. Summary of Data Sources and Variables.**
(DOCX)

**S2 Table.** Age-adjusted SARS-CoV-2 seroprevalence among LTCH/RH staff. Study figures, R packages used in the methods, code, and figure codes are provided at: https://github.com/mishra-lab/COVID-WellnessHubSeroprevalence.git.
(DOCX)

## Acknowledgments

We acknowledge Melanie Delgado-Brand, Tulunay Tursun, Geneviève Mailhot, Martina Tersigni, Adrian Pasculescu, and Freda Qi of the Gingras laboratory for their role in ELISA analysis. Anne-Claude Gingras leads the Functional Genomics and Structure-Function Pillar of CoVaRR-Net. Antigens, protein standards and secondary antibodies for ELISA were kindly provided by The Pandemic Response Challenge Program of the National Research Council of Canada (Dr. Yves Durocher); positive and negative control samples for ELISA assay calibration were from the National Microbiology Laboratory, Public Health Agency of Canada (Dr. John Kim). The robotics equipment used is housed in the Network Biology Collaborative Centre at the Lunenfeld-Tanenbaum Research Institute, a facility supported by the Canada Foundation for Innovation, the Ontario Government, and Genome Canada and Ontario Genomics (OGI-139). We thank Linwei Wang (Unity Health Toronto) for R scripts that were adapted to generate Fig 1. Finally, we would like to thank all of the organizational partners of the Wellness Hub program for their support of this study. A full list of partner organizations can be found at https://wellness-hub.ca/. The authors thank Andreea Manea and Negin Pak for supporting manuscript preparation.

## Author contributions

**Conceptualization:** Christine Fahim, Jamie M. Boyd, Huiting Ma, Keelia Quinn de Launay, Alyson Takaoka, Anne-Claude Gingras, Sharon E. Straus, Sharmistha Mishra.

**Data curation:** Christine Fahim, Nimitha Paul, Vincenza Gruppuso, Ana Mrazovac, Jessica Firman, Anjali Patel, Keelia Quinn de Launay, Alyson Takaoka, Vanja Grubac.

**Formal analysis:** Christine Fahim, Siyi Wang, Nimitha Paul, Karen Colwill, Roya Dayam, Huiting Ma, Anne-Claude Gingras, Sharmistha Mishra.

**Funding acquisition:** Christine Fahim, Anne-Claude Gingras, Sharon E. Straus, Sharmistha Mishra.

**Investigation:** Keelia Quinn de Launay, Vanja Grubac.

**Methodology:** Christine Fahim, Siyi Wang, Karen Colwill, Roya Dayam, Jamie M. Boyd, Huiting Ma, Anne-Claude Gingras, Sharon E. Straus, Sharmistha Mishra.

**Project administration:** Christine Fahim, Nimitha Paul, Jamie M. Boyd, Vincenza Gruppuso, Ana Mrazovac, Jessica Firman, Anjali Patel, Vanessa Bach, Keelia Quinn de Launay, Alyson Takaoka, Vanja Grubac.

**Resources:** Anne-Claude Gingras.

**Software:** Siyi Wang, Nimitha Paul, Roya Dayam, Huiting Ma, Sharmistha Mishra.

**Supervision:** Christine Fahim, Karen Colwill, Jamie M. Boyd, Huiting Ma, Keelia Quinn de Launay, Anne-Claude Gingras, Sharon E. Straus, Sharmistha Mishra.

**Validation:** Siyi Wang, Nimitha Paul, Vincenza Gruppuso, Ana Mrazovac, Jessica Firman, Anjali Patel, Vanessa Bach, Anne-Claude Gingras, Sharmistha Mishra.

**Visualization:** Sharmistha Mishra.

**Writing – original draft:** Christine Fahim, Siyi Wang, Karen Colwill, Roya Dayam, Sharon E. Straus, Sharmistha Mishra.

**Writing – review & editing:** Christine Fahim, Siyi Wang, Nimitha Paul, Karen Colwill, Roya Dayam, Jamie M. Boyd, Huiting Ma, Vincenza Gruppuso, Ana Mrazovac, Jessica Firman, Anjali Patel, Vanessa Bach, Keelia Quinn de Launay, Alyson Takaoka, Vanja Grubac, Anne-Claude Gingras, Sharon E. Straus, Sharmistha Mishra.

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
