## [Decision Letter · Decision Letter 0]

29 Nov 2024

PGPH-D-24-01292

Patterns of SARS-CoV-2 seropositivity among essential workers in long term care and retirement homes in Ontario, Canada: A descriptive cross-sectional study

Dear Dr. Fahim,

Thank you for submitting your manuscript to PLOS Global Public Health. After careful consideration, we feel that it has merit but does not fully meet PLOS Global Public Health’s publication criteria as it currently stands. Therefore, we invite you to submit a revised version of the manuscript that addresses the points raised during the review process.

Your manuscript has been assessed by two reviewers and their comments are available below. The reviewers have requested some additional detail in the methodology as well and more clearly defined conclusions in relation to the studies aims and hypotheses. Please review their comments and make the appropriate revisions. 

We look forward to receiving your revised manuscript.

Kind regards,

Emma Campbell, Ph.D

Staff Editor

Journal Requirements:

**Please only choose the relevant sentences from below**

1) Please clarify all sources of funding (financial or material support) for your study. List the grants (with grant number) or organizations (with url) that supported your study, including funding received from your institution. 

2) State the initials, alongside each funding source, of each author to receive each grant.

3) State what role the funders took in the study. If the funders had no role in your study, please state: “The funders had no role in study design, data collection and analysis, decision to publish, or preparation of the manuscript.”

4) If any authors received a salary from any of your funders, please state which authors and which funders.

3. Please provide an Author Summary. This should appear in your manuscript between the Abstract (if applicable) and the Introduction, and should be 150–200 words long. The aim should be to make your findings accessible to a wide audience that includes both scientists and non-scientists. Sample summaries can be found on our website under Submission Guidelines: 

https://journals.plos.org/globalpublichealth/s/submission-guidelines#loc-parts-of-a-submission

4. Please provide separate figure files in .tif or .eps format.

5. We have noticed that you have uploaded Supporting Information files, but you have not included a list of legends. Please add a full list of legends for your Supporting Information files after the references list. 

Additional Editor Comments (if provided):

Reviewers' comments:

Reviewer's Responses to Questions

**Comments to the Author**

1. Does this manuscript meet PLOS Global Public Health’s publication criteria ? Is the manuscript technically sound, and do the data support the conclusions? The manuscript must describe methodologically and ethically rigorous research with conclusions that are appropriately drawn based on the data presented.

Reviewer #1: Yes

Reviewer #2: Yes

2. Has the statistical analysis been performed appropriately and rigorously?

Reviewer #1: Yes

Reviewer #2: Yes

3. Have the authors made all data underlying the findings in their manuscript fully available (please refer to the Data Availability Statement at the start of the manuscript PDF file)?

Reviewer #1: Yes

Reviewer #2: Yes

4. Is the manuscript presented in an intelligible fashion and written in standard English?

Reviewer #1: Yes

Reviewer #2: Yes

5. Review Comments to the Author

Reviewer #1: This study used purposive and convenient sample to understand the correlation between the socio-economic factors and variability of SARS-CoV-2 sero-prevalence. Socioeconomic disparities are found to be strongly associated with the sero-prevalence variability. Well-written manuscript and authors reported the limitations due to data.

Minor comment: Please list R packages used in the methods, and share the analyses code (without data) and figure codes via github repository. This will help research community.

Reviewer #2: 1. SARS-Cov-2 infection is not classified as seroprevalence. Authors must look at the again. In the abstract, line 50 authors stated that "Positive seroprevalence was defined as ....". Authors should state what positive SARS-CoV-2 infection is and what seroprevalence was defined as separately.

2. Generally the manuscript was well written and the data analysed well

3. The conclusion does not support the objectives set for the study. This needs to be rewritten.

4. References should be written according to the journal guidelines

6. PLOS authors have the option to publish the peer review history of their article (what does this mean? ). If published, this will include your full peer review and any attached files.

**Do you want your identity to be public for this peer review?** For information about this choice, including consent withdrawal, please see our Privacy Policy .

Reviewer #1: No

Reviewer #2: No

---

## [Decision Letter · Decision Letter 1]

29 Jan 2025

Patterns of SARS-CoV-2 seropositivity among essential workers in long term care and retirement homes in Ontario, Canada: A descriptive cross-sectional study

PGPH-D-24-01292R1

Dear Dr Fahim,

We are pleased to inform you that your manuscript 'Patterns of SARS-CoV-2 seropositivity among essential workers in long term care and retirement homes in Ontario, Canada: A descriptive cross-sectional study' has been provisionally accepted for publication in PLOS Global Public Health.

Best regards,

Raquel Muñiz-Salazar, Ph.D.

Academic Editor

We are pleased to inform you that your manuscript, titled Patterns of SARS-CoV-2 Seropositivity among Essential Workers in Long Term Care and Retirement Homes in Ontario, Canada: A Descriptive Cross-Sectional Study, has been formally accepted for publication in PLOS Global Public Health.

We appreciate your prompt and comprehensive responses to the reviewers' comments, which have significantly enhanced the clarity and rigor of the manuscript.

Reviewer Comments (if any, and for reference):

Reviewer's Responses to Questions

**Comments to the Author**

1. If the authors have adequately addressed your comments raised in a previous round of review and you feel that this manuscript is now acceptable for publication, you may indicate that here to bypass the “Comments to the Author” section, enter your conflict of interest statement in the “Confidential to Editor” section, and submit your "Accept" recommendation.

Reviewer #2: All comments have been addressed

2. Does this manuscript meet PLOS Global Public Health’s publication criteria ? Is the manuscript technically sound, and do the data support the conclusions? The manuscript must describe methodologically and ethically rigorous research with conclusions that are appropriately drawn based on the data presented.

Reviewer #2: Yes

3. Has the statistical analysis been performed appropriately and rigorously?

Reviewer #2: Yes

4. Have the authors made all data underlying the findings in their manuscript fully available (please refer to the Data Availability Statement at the start of the manuscript PDF file)?

Reviewer #2: Yes

5. Is the manuscript presented in an intelligible fashion and written in standard English?

Reviewer #2: Yes

6. Review Comments to the Author

Reviewer #2: All my comments have been duly addressed by the authors

7. PLOS authors have the option to publish the peer review history of their article (what does this mean? ). If published, this will include your full peer review and any attached files.

**Do you want your identity to be public for this peer review?** For information about this choice, including consent withdrawal, please see our Privacy Policy .

Reviewer #2: No
